# DISTRIBUTED TRAINING OF GRAPH CONVOLUTIONAL NETWORKS USING SUBGRAPH APPROXIMATION

## ABSTRACT

Modern machine learning techniques are successfully being adapted to data modeled as graphs. However, many real-world graphs are typically very large and do not fit in memory, often making the problem of training machine learning models on them intractable. Distributed training has been successfully employed to alleviate memory problems and speed up training in machine learning domains in which the input data is assumed to be independently identical distributed (i.i.d). However, distributing the training of non i.i.d data such as graphs that are used as training inputs in Graph Convolutional Networks (GCNs) causes accuracy problems since information is lost at the graph partitioning boundaries.

In this paper, we propose a training strategy that mitigates the lost information across multiple partitions of a graph through a subgraph approximation scheme. Our proposed approach augments each sub-graph with a small amount of edge and vertex information that is approximated from all other sub-graphs. The subgraph approximation approach helps the distributed training system converge at single-machine accuracy, while keeping the memory footprint low and minimizing synchronization overhead between the machines.

## 1 INTRODUCTION

Graphs are used to model data in a diverse range of applications such as social networks (Hamilton et al., 2017a), biological networks (Fout et al., 2017) and e-commerce interactions (Yang, 2019). Recently, there has been an increased interest in applying machine learning techniques to graph data. A breakthrough was the development of Graph Convolutional Networks (GCNs), which generalize the function of Convolutional Neural Networks (CNNs), to operate on graphs (Kipf & Welling, 2017).

Training GCNs, much like training CNNs, is a memory and computationally demanding task and may take days or weeks to train on large graphs. GCNs are particularly memory demanding since processing a single vertex may require accessing a large number of neighbors. If the GCN algorithm accesses neighbors which are multiple hops away, it may need to traverse disparate sections of the graph with no locality. This requirement poses challenges when the graph does not fit in memory, which is often the case for large graph representations. Previous approaches to alleviate issues of memory and performance seeks to constrain the number of dependent vertices by employing sampling-based methods (Hamilton et al., 2017b; Chen et al., 2018; Huang et al., 2018; Chiang et al., 2019). However, sampling the graph may limit the achievable accuracy (Jia et al., 2020).

CNN training turned to data parallel distributed training to alleviate the memory size constraints and the computing capacity of a single machine. Here, a number of machines independently train the same neural network model but on an assigned chunk of the input data. To construct a single global model, the gradients or parameters are aggregated at regular intervals, such as at the completion of a batch (Dean et al., 2012). While this approach works well for CNNs, it is more challenging for GCNs due to the nature of the input data. If the training input is a set of images, each image is independent. Hence, when the dataset is split into chunks of images, no information is directly carried between the chunks. In contrast, if the input is a graph, the equivalent of splitting the dataset into chunks is to divide the large graph into a number of subgraphs. However, GCNs train on graphs which rely not only on information inherent to a vertex but also information from its neighbors.

When a GCN trains on one subgraph and needs information from a neighbor in another subgraph, the information from the edges that span across machines creates a communication bottleneck. For

example, in our experimental evaluations we note that when the Reddit dataset (Hamilton et al., 2017b) is partitioned into five subgraphs, there are about 1.3 million (bidirectional) edges that span across machines. If each edge is followed, 2.6 million messages would be sent, each containing a feature vector of 602 single-precision floats, summing up to about 6.3 GB of data transferred each epoch, where one epoch traverses the entire graph. Each training typically requires many epochs. In our experiments running distributed training on the GCN proposed by Kipf & Welling (2017) one epoch takes about one second. Thus the bandwidth required is enormous. This is a serious scalability issue: since the time it takes to finish one epoch decreases when the training is scaled out on more machines, the required bandwidth increases. Even with optimizations such as caching that might be possible (e.g. vertex caching (Yang, 2019)), the amount of communication is significant.

Alternatively, instead of communicating between subgraphs, a GCN may ignore any information that spans multiple machines (Jia et al., 2020). However, we observe that this approach results in accuracy loss due to suboptimal training. For example, when distributed on five compute nodes and ignoring edges that span across machines, we note in our evaluations that the accuracy of GraphSAGE trained on the Reddit dataset drops from 94.94% to 89.95%, a substantial loss in accuracy.

To mitigate this problem, we present a distributed training approach of GCNs which reaches an accuracy comparable with training on a single-machine system, while keeping the communication overhead low. This allows for GCNs to train on very large graphs using a distributed system where each machine has limited memory, since the memory usage on each machine scales with the size of its allotted (local) subgraph. We achieve this by approximating the non-local subgraphs and making the approximations available in the local machine. We show that, by incurring a small vertex overhead in each local machine, a high accuracy can be achieved. Our contributions in this paper are:

- A novel subgraph approximation scheme to enable distributed training of GCNs on large graphs using memory-constrained machines, while reaching an accuracy comparable to that achieved on a single machine.
- An evaluation of the approach on two GCNs and two datasets, showing that it considerably increases convergence accuracy, and allows for faster training while keeping the memory overhead low.

The rest of the paper is organized as follows. Section 2 walks through the necessary background and presents experimental data that motivates our approach. Section 3 describes our approach to efficiently distribute GCNs. Section 4 describes our evaluation methodology, while Section 5 presents the results of our evaluation. Section 6 discusses related work before we conclude in Section 7.

## 2 Background and Motivation

In this section we first describe GCNs. Then, we show that GCNs are resilient to approximations in the input graph, and describe how that motivates our approach in Section 3.

### 2.1 Graph Convolutional Networks

GCNs are neural networks that operate directly on graphs as the input data structure. A GCN aims to define an operation similar to a convolutional layer in CNNs, where a parameterized filter is applied to a pixel and its neighbors in an image (Fig 1). In the graph setting, the pixel corresponds to a vertex, and the edges from the vertex point out its neighbors (Fig 2). Early approaches to define this operation used graph convolutions from spectral graph theory (Bruna et al., 2014; Henaff et al., 2015). While theoretically sound, these approaches are difficult to scale to large graphs. (Wu et al., 2020). Recent research (e.g. (Hamilton et al., 2017b; Chen et al., 2018)) instead takes a spatial-based approach, which enables computation in batches of vertices.

The goal of a GCN is to learn the parameters of a neural network given a graph $G = (V, E)$ where $V$ is a set of vertices and $E$ is a set of edges. The input is a feature matrix $X$, where each row represents the feature vector $x_v$ for every vertex $v \in V$, and an adjacency matrix $A$ which represents the edges $e \in E$. Each layer in can be described as $H^i = f(H^{i-1}, A), H^0 = X$, where $i$ is the layer order and $f$ is a propagation rule (Kipf & Welling, 2017). The number of layers and the propagation rule differ between GCNs. While our proposal is general to all GCNs of this type, we evaluate it on the GCN proposed by Kipf & Welling (2017), which we call KW-GCN, and GraphSAGE proposed by Hamilton et al. (2017b). Both GCNs are described in more detail in Section 4.1.

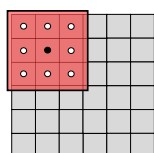 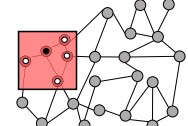 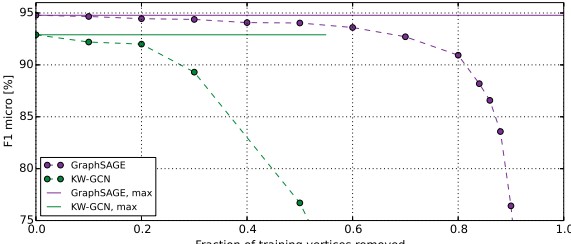

Figure 1: Image convolution.

Figure 2: Graph convolution.

Figure 3: Accuracy when randomly removing a certain fraction of the training vertices.

## 2.2 LIMITATIONS TO INPUT APPROXIMATIONS

Our proposal builds upon the observation that GCNs may be resilient to approximations in the input graph. To investigate the extent of resilience, we trained both KW-GCN and GraphSAGE on the Reddit dataset (Hamilton et al., 2017b), described in Section 4.2, for the task of vertex classification.

To introduce approximations in the input graph, we randomly select an increasingly larger fraction of the training vertices to be deleted. We then use the approximated graph to train each GCN and compute the micro F1 score (the micro-averaged harmonic mean of precision and recall, which we call *accuracy*) on the complete test set. Figure 3 shows the results. We make the following qualitative observation: while GraphSAGE and KW-GCN are not equivalent in their resiliency towards vertex deletion, both networks demonstrate that up until a certain point it is possible to delete vertices without a large accuracy drop. While this might seem like a trivial but promising approach to train more efficiently by simply dropping vertices, in practice the threshold for deletion must be large to make a significant impact on the memory footprint: 50% of the vertices has to be removed to reduce the memory footprints by 2X. Such a high deletion fraction incurs an unacceptable accuracy penalty.

In this work, we instead leverage this observation to gain partial knowledge about non-local subgraphs in a distributed setting. While this approach increases the total workload by introducing redundancy, we show in Section 5 that only a small overhead is needed to substantially increase accuracy. In the next section we describe our proposed solution that exploits the approximation resilience property.

## 3  DISTRIBUTED TRAINING OF GCNS USING SUBGRAPH APPROXIMATION

A common approach to achieve distributed training is to leverage data parallelism: the training data is divided into mutually exclusive sets, and each machine is assigned one of these sets. Aside from decreasing training time, data parallelism also reduces memory footprint by lowering the amount of data each machine has to handle. If the graph does not fit in memory, the performance will be significantly impacted due to storage access time (Ham et al., 2016; Ahn et al., 2015).

The main challenge of distributed subgraph training is accuracy. Each machine has only knowledge about the graph structure in its own partition. The result is that important information embedded in other parts of the graph is not captured in local subgraphs. Therefore, the local neural network may not learn global patterns. This is a unique problem to graph data structures.

If the input to the neural network consists of images, no information is lost when the dataset is distributed, since each image is an independent data element. If the input is a graph, some edges that span the partitions are essentially unaccounted for during training. If the convolution operator (Figure 2) targets a vertex at the edge of the subgraph, the neighbor data needed is not available in the local machine. A choice has to be made: either acknowledge the edge by trying to fetch the missing data from the non-local machine, or ignore the edge. Both approaches introduce inefficiencies.

If the edge is acknowledged, there is a risk of overflowing the network with communication calls, since the machine that holds the neighboring vertex first has to be identified, and then that machine has to send out its data. The number of communication calls grows with the number of machines. However, if the edge is ignored, the execution time of each machine decreases since the communication overhead is minimized, but information is lost: the convolution will be approximate.

---

**Algorithm 1:** Breadth-first random sampling of vertices

---

**Input:** $N_p$: number of vertices in partition p; M: number of partitions; P: partitions; O: overlap
**Output:** $P_o$: Partitions with overlap O

1 **for** *p in P* **do**
2     nts = $\frac{O \cdot N_p}{M-1}$; $P_o[p]$ = p
3     **for** *op in $P \setminus p$* **do**
4        sample($P_o[p]$, p, op, nts)

5 **Function** *sample($P_o[p]$, p, op, nts)*:
6     selection = neighbors(p, op)
7     **if** *size of selection >= nts* **then**
8        sn = random(selection, nts); $P_o[p]$ = sn $\cup$ $P_o[p]$
9     **else**
10        $P_o[p]$ = selection $\cup$ $P_o[p]$
11        hop = neighbors(selection, op) $\setminus$ selection
12        **if** *! hop is empty* **then** sample($P_o[p]$, hop, op, nts - size of selection) ;

---

### 3.1 SUBGRAPH APPROXIMATION

We overcome the problem of information loss by approximating the subgraphs which are not locally available. The intuition behind this approach is that each machine will have a complete view of one partition, and an *approximate view* of the rest of the partitions.

After partitioning the graph into a number of subgraphs, the following steps are taken for each subgraph $g_i$: 1) A fixed number of vertices are randomly sampled from each of the other subgraphs, if the vertex has a path to $g_i$. 2) The subgraph $g_i$ is extended to include these vertices from other sub-graphs thereby creating an approximate view of all the sub-graphs that are on other machines.

While there are multiple approaches to sampling, in this work we use a breadth-first algorithm as shown in Algorithm 1. First, we calculate how many vertices we need to sample ($nts$) from the other partitions (line 2) based on an overlap threshold $O$. $O$ can be determined by the user based on the additional memory overhead that the system is willing to accommodate. For example, when distributed across five machines, a 10% overlap threshold means that each partition can hold 2.5% additional vertices from all the other four sub-graphs ($4X2.5\%$), compared to the vertex count in its current partition. To select vertices, our algorithm first finds all the one-hop neighbors in the other partition (line 6). If the selection exceeds the number of vertices we wish to sample, we randomly select $nts$ vertices from the selection and include them in the new partition (line 8).

If the number of one-hop neighbors is smaller than $nts$, we consider vertices farther away. First, we add the current-hop neighbors to the new partition (line 10), and then find the next-hop neighbors (line 11). We then recursively sample next-hop neighbors (line 12) until $O$ is met. We estimate the worst-case complexity of the algorithm to be $\mathcal{O}(N^2)$, were $N$ is the number of vertices in the graph.[1]

### 3.2 TRAINING WITH SUBGRAPH APPROXIMATION

The training process with subgraph approximation uses a master-worker setting. The *master* acts as a parameter server, and is responsible for initiating the training by partitioning the graph, applying subgraph approximation, and distributing the partitions among the workers.

The *workers* are responsible for the training. Each worker $i$ has a local copy $W$ of the neural network, and trains on its assigned subgraph. At the end of every epoch $k$, the worker sends its trained parameters $W_i^k$ to the master. The master collects the parameters and averages them before distributing the result, $W_{agg}^k$, back to the workers. After receiving the averaged parameters, each worker continues the training on its own subgraph for the next epoch. At the end of the training period, the aggregated parameters from each of the workers form a global model.

---

[1]For derivation, refer to Appendix A.

Table 1: Dataset Statistics

| Dataset | #Vertices | #Edges | #Labels | #Features | % of vertices in Train/Val/Test |
|---------|-----------|--------|---------|-----------|----------------------------------|
| Reddit | 231,443 | 11,606,919 | 41 | 602 | 70/20/10 |
| Amazon2M | 2,683,902 | 48,336,954 | 31 | 100 | 70/20/10 |

During each epoch the vertices at the frontier of each subgraph use the approximated neighbors to capture the neighborhood information. Recall that during training the vertices at the frontier in each subgraph only need to capture information from their immediate neighbors (1 or 2-hop neighbors at most). Our approximate subgraphs capture this information. In the next epoch this information from one subgraph must move deeper into other subgraphs which are not present on the local machine. This is where the parameter aggregation at the end of an epoch plays a role in communicating knowledge across subgraphs. Thus intuitively we capture both the local knowledge at each subgraph boundary during an epoch, and then transmit that knowledge across subgraphs between epochs.

## 4    EVALUATION METHODOLOGY

We evaluate our approach on a cluster of identical machines, each equipped with an AMD Ryzen 3 1200 Quad-Core processor, 32 GB memory, and a NVIDIA GeForce GTX 1080 Ti. The cluster is configured as described in Section 3.2. For graph partitioning we use the METIS library (Karypis & Kumar, 1997), choosing edge cut as the minimization objective. Each subgraph has approximately equal number of vertices. After including a sampling of edges from other sub-graphs using Algorithm 1 all other edges between the subgraphs are dropped.

### 4.1    INVESTIGATED GCNS

We implement distributed versions of the KW-GCN and GraphSAGE in PyTorch (Paszke et al., 2017), and apply them in an inductive setting. We give a brief overview of the GCNs and a specification of the applied hyperparameters. For an in-depth description we refer the reader to the original papers.

KW-GCN builds on an approximation of spectral graph convolutions (Kipf & Welling, 2017). The propagation rule is defined as $H^L = \sigma(\hat{A}H^{L-1}W^L), H^0 = X$.[2] For evaluation we use the Adam (Kingma & Ba, 2014) optimizer, 128 hidden dimensions, a learning rate of 0.02, and 2 layers.

GraphSAGE employs a spatial-based approach which uses sampling to aggregate feature information. It learns in three steps: 1) Sample a fixed number of neighbors. 2) Derive the vertex embedding by aggregating its neighbors' feature information. 3) Use the embedding to predict the graph label. We use the element-wise mean as propagation rule together with 2 layers, a batch-size of 256, the Adam optimizer, and learning rates of 0.0001 (Reddit) and 0.0002 (Amazon2M).

### 4.2    DATASETS

We evaluate our approach by classifying vertices in the Reddit and Amazon2M datasets (Table 1).

The Reddit dataset (Hamilton et al., 2017b) is a graph representation of Reddit posts made during one month, where each vertex in the graph is a post. There is an edge between two posts if the same user commented on them both. The vertex label is the community a post belongs to. The features consist of an embedding of post information, created using GloVe CommonCrawl (Pennington et al., 2014). The first 20 days of posts are used for training, while the rest are used for testing and validation.

The Amazon2M dataset presented by Chiang et al. (2019) represents purchases where each vertex is an item and an edge represents that the items were bought together. Our graph is based on the same raw data, but we extract 100-dimensional vertex features from the item descriptions using Word2Vec (Mikolov et al., 2013) instead of bag-of-words. In the raw data, each item has many categories. We assign labels by counting the categories and selecting the most-frequently occurring category as the label. The vertices in the dataset split are randomly selected.

---

[2] $\hat{A}$: normalized adjacency matrix, $X$: feature matrix, $W^L$: trainable weight matrix for layer $L$

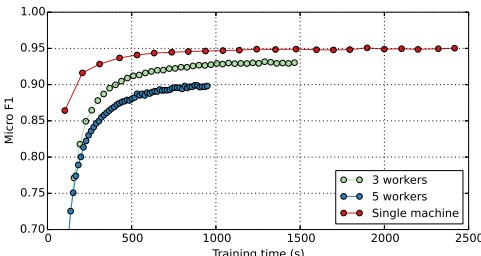 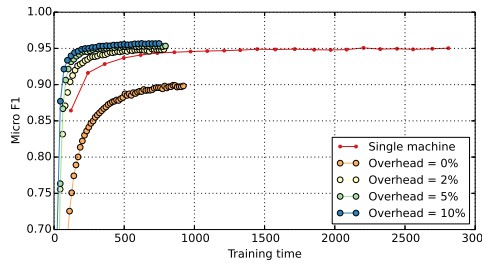

Figure 4: Training GraphSAGE with the Reddit dataset *without* subgraph approximation.

Figure 5: Training GraphSAGE on 5 workers with the Reddit dataset, using subgraph approximation.

## 5 RESULTS

### 5.1 IMPACT ON CONVERGENCE

Figure 5.4 shows an example of training GraphSAGE on the Reddit dataset using both a single-machine and a distributed approach but *without* using subgraph approximation. In the latter approach, the graph is partitioned into mutually exclusive subgraphs. We let the model train until convergence.

When training the model using the full graph on a single machine, the model converges at an accuracy of 94.89%. However, when training in a distributed setting without subgraph approximation technique the accuracy converges at 93.09 and 89.88% for 3 and 5 workers respectively. When the number of partitions increase, the model converges at an increasingly lower accuracy. This behaviour is due to the reasons we described in Section 3, i.e. that each machine only has a local view of the graph, and that dropping edges creates a loss of information about neighbouring vertices in other subgraphs.

### 5.2 IMPACT OF SUBGRAPH APPROXIMATION

Figure 5 shows an example of how subgraph approximation can increase both accuracy and training time of distributed GCN training. We use five workers and let the model train until convergence using four overlap thresholds ($O$ in Algorithm 1): 0%, 10%, 25%, and 50%. These overlaps correspond to an overhead of 0%, 2%, 5%, and 10% of the *total* number of vertices in each partition.

Without overhead, the model converges at a substantially lower accuracy than the single-machine case. In contrast, an overhead of 2% (10% overlap) increases the convergence rate and allows the model to converge to single-machine accuracy. A further overhead increase helps the model to converge even faster but does not necessarily increase accuracy. We now provide more details from each dataset.

*Subgraph Approximation on the Reddit Dataset*

Table 2 summarizes results for training KW-GCN and GraphSAGE using the Reddit dataset on three and five workers. When training GraphSAGE on three workers, the accuracy converges at about 2 percentage units below the single-machine case without any overhead. However, when adding an overhead of 3.3% (10% overlap) in each worker, the model reaches single-machine accuracy.

We see a similar behaviour when training on five workers: a small overhead of 2% (10% overlap) considerably improves the accuracy. However, an overhead slightly larger than 2% is needed to reach single-machine accuracy. We notice that as the number of workers increase, the overhead necessary to reach single machine accuracy increases slightly but not substantially.

The speedup for each configuration is reported in Table 2. Note that speedup is sub-linear with the number of machines. This is because the parameters must be aggregated at the end of each epoch which is a serialization bottleneck. Our machines are connected with 1Gbps Ethernet links due to cost limitations, which acts as scalability barriers. We believe that using e.g. Infiniband will improve the scalability, but even within the limited cost we were able to achieve about 2X speedup.

As observed in Figure 3, KW-GCN is less resilient to input approximations than GraphSAGE is. This is reflected in the distributed behaviour of KW-GCN. As Table 2 shows, KW-GCN is more sensitive to partitioning: with no overhead the accuracy converges at about 0.27 (three workers) and 0.42 (five

Table 2: Results for training GraphSAGE (GS) and KW-GCN (KW) using Reddit.

| | Three workers | | | | | | | |
| | Accuracy [Micro F1] | | Training Time [s] | | Speedup | | Allocated GPU Memory [GB][3] | |
| | KW | GS | KW | GS | KW | GS | KW | GS |
|---|---|---|---|---|---|---|---|---|
| Single machine | 92.90 | 94.94 | 784 | 2056 | - | - | 1.86 | 2.79 |
| 0% overhead | 65.13 | 93.12 | 376 | 1182 | 2.08 | 1.74 | 0.53-0.68 | 0.95-1.0 |
| 3.3% overhead | 91.87 | 95.50 | 406 | 1035 | 1.93 | 1.99 | 0.58-0.72 | 1.26-1.27 |
| 8.33% overhead | 92.27 | 95.65 | 453 | 1144 | 1.73 | 1.80 | 0.65-0.78 | 1.50-1.53 |
| 16.67% overhead | 92.71 | 95.61 | 528 | 1098 | 1.48 | 1.87 | 0.82-0.93 | 1.86-1.89 |
| | Five workers | | | | | | | |
| | Accuracy [Micro F1] | | Training Time [s] | | Speedup | | Allocated GPU Memory [GB] | |
| | KW | GS | KW | GS | KW | GS | KW | GS |
| Single machine | 92.90 | 94.94 | 784 | 2056 | - | - | 1.86 | 2.79 |
| 0% overhead | 50.20 | 89.95 | 333 | 991 | 2.35 | 2.07 | 0.31-0.41 | 0.51-0.55 |
| 2% overhead | 88.88 | 94.87 | 368 | 889 | 2.13 | 2.31 | 0.33-0.44 | 0.57-0.62 |
| 5% overhead | 91.80 | 95.35 | 385 | 735 | 2.03 | 2.80 | 0.37-0.42 | 0.71-0.76 |
| 10% overhead | 91.93 | 95.63 | 427 | 758 | 1.83 | 2.71 | 0.45-0.51 | 0.97-0.99 |

*Note: All results are averaged over multiple runs.*

Table 3: Training KW-GCN on Amazon2M until convergence on five workers.

| Overhead | Micro F1 | Time | Alloc. GPU Memory |
|---|---|---|---|
| 0% | 78.40 | 1022s | 1.83-2.31 GB |
| 2% | 79.10 | 1148s | 2.06-2.52 GB |
| 5% | 79.50 | 1290s | 2.39-2.52 GB |
| 10% | 79.90 | 1566s | 2.97-3.43 GB |

*Note: All results are averaged over multiple runs.*

Table 4: Training GraphSAGE on Amazon2M for a fixed time (2386 s) on 20 workers.

| Overhead | Micro F1 | Epochs | Alloc. GPU Memory |
|---|---|---|---|
| 0% | 85.55 | 80 | 0.89 GB |
| 0.5% | 86.60 | 63 | 0.90 GB |
| 1.25% | 86.89 | 53 | 0.93 GB |
| 2.50% | 87.01 | 40 | 0.98 GB |

*Note: All results are averaged over multiple runs.*

workers) units below the single-machine baseline, which is an unacceptable loss. However, adding a small overhead substantially increases the model accuracy bringing it very close to the baseline.

*Subgraph Approximation on the Amazon2M Dataset*

To explore how subgraph approximation scales with larger datasets we also investigate the Amazon2M dataset, which has about 10 times more vertices than the Reddit dataset. This dataset is challenging to train on a single machine for both GCNs, for different reasons: for KW-GCN, the adjacency matrix is too large to fit in GPU memory. For GraphSAGE, the training time is very long.

For KW-GCN, distribution of Amazon2M is not just an optional exercise but it is critical for training to even start. All our graphs were able to fit within the system memory limitations once they are divided into at least three subgraphs. For instance, the Amazon2M dataset was unable to fit in any single GPU memory but when sub-divided into five subgraphs it was able to fit across all the five GPU memories. Hence, at least it is now feasible to train Amazon2M on a reasonably priced GPU. Table 3 shows the result of training KW-GCN on five workers. While it's not possible to train on one machine, the distributed approach enables training of KW-GCN. Furthermore, subgraph approximation significantly increases the accuracy: a 2% overhead results in 1.5% higher accuracy.

We also train GraphSAGE on the Amazon2M dataset, distributed on 20 workers. Adding an overhead increases the per-epoch execution time. However, this increase in traverse time is worthwhile, since fewer epochs are needed to reach a higher accuracy. We show this in Table 4: here, each overhead configuration has been trained for 2386 seconds, which is the time it takes to train the non-overhead configuration for 80 epochs. The accuracy increases significantly with an added overhead, although the number of epochs decrease: a small overhead of 2.5% increases accuracy from 85.55% to 87.01%.

## 5.3 DISCUSSION

In all GCN and dataset combinations, we see a substantial increase in accuracy when adding a small overhead. This behaviour is due to 1) the number of GCN layers, and 2) the graph density.

When it comes to property 1), for both KW-GCN and GraphSAGE we use two layers, which means that they aggregate information from neighbors two hops away. When adding a single vertex from a

---

[2]All memory measurements are retrieved with `torch.cuda.allocated()`

non-local partition, it's not only the one-hop neighbors that benefit, but also all two-hop neighbors. Hence, the importance of each added vertex is super-linearly beneficial. However, the importance is less pronounced as the overlap grows: at some point, each vertex has enough information of its neighbors for the neural network to be able to learn from that vertex.

Another reason for the benefit from overlapped sub-graphs is the vertex degree. Reddit and Amazon2M are datasets with a relatively high average vertex degree (100.03 and 36.02, respectively). Naturally, if the added vertex has many edges to the local subgraph, the information from that single vertex will be beneficial to many vertices in the local subgraph. Hence, datasets with reasonable vertex degree are more likely to derive super-linear benefits from adding just a few vertices.

### 5.4 CONVERGENCE RATE ANALYSIS

To compare the convergence rate of our distributed solution with ClusterGCN which allows large-graph training on a single machine, we trained ClusterGCN implemented in PyG (Fey & Lenssen, 2019) using the hyperparameter setting from the original paper. Our goal is not to compare execution time, which depends on the implementation, but rather to study the convergence rate, which depends on the number of epochs needed to reach convergence and the per-epoch training time. We find experimentally that ClusterGCN and GraphSAGE distributed on 5 machines with 2% overhead converges in about the same number of epochs. Hence, the per-epoch time decides convergence rate.

To compare per-epoch training time, we investigate the time complexity for the two algorithms. According to the original paper, time complexity for ClusterGCN for an $N$ node graph with $F$ features is $\mathcal{O}(L||A0||F + LNF^2)$, and $\mathcal{O}(r^L NF^2)$ for single-machine GraphSAGE[4]. Assuming $L$ and $r$ are small constants, we simplify to $\mathcal{O}(||A0||F + NF^2)$ and $\mathcal{O}(NF^2)$.

Given the above, the time complexity of distributed GraphSAGE is $\mathcal{O}(\frac{(N \cdot O)F^2}{M})$, where $M$ is the number of machines and $O$ is the overlap from subgraph approximation. Given that $O$ can be held low as we have shown previously ($< 1.2$ according to Section 5.2), our conclusion is that distributed GraphSAGE converges faster than ClusterGCN if they're implemented in the same framework.

## 6 RELATED WORK

The importance of distributing the training of neural networks has driven a large body of work. The crucial aspect of efficiently sharing parameters across the distributed models has been rigorously investigated, for example in terms of compression (Yu et al., 2018; Alistarh et al., 2017; Dryden et al., 2016; Wen et al., 2017; Lin et al., 2018) and parallel aggregation (Thakur et al., 2005; Goyal et al., 2017). Optimization of parameter sharing is relevant also in the setting of a distributed GCN, but it is orthogonal to the problem we aim to mitigate (communication overhead caused by the input graph).

NeuGraph (Ma et al., 2019) and Cluster-GCN (Chiang et al., 2019) are frameworks which enable parallel GNN training on a *single* machine. However, they do not consider a fully distributed setting. Furthermore, they assume that the full graph is readily available, which is not the case in a distributed setting. Since they do not target accuracy loss inflicted by lost edges, their approaches are orthogonal.

Aligraph (Yang, 2019) and Roc (Jia et al., 2020) are both frameworks which targets distributed GNN training. Aligraph communicates all neighbor information between the subgraphs, while Roc trains on isolated subgraphs. Subgraph approximation can be applied to further improve these frameworks by removing unnecessary communication (Aligraph) and increase accuracy (Roc).

## 7 CONCLUSION

While distributing the training of GCNs is an efficient approach to reduce execution time and memory footprint, it also causes accuracy problems since information is lost between the graph partitions. In this paper, we propose a technique for mitigating the lost information by introducing subgraph approximation. By letting the subgraphs overlap slightly, the system can reach state-of-the-art accuracy while still keeping the memory footprint low and at the same time avoiding costly synchronization caused by acknowledging edges between subgraphs placed on different machines.

---

[4] $L$ = #layers, $||A0||$ = #non-zeros in adjacency matrix, $r$ = #sampled neighbors

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

## A   APPENDIX: SUPPLEMENTARY MATERIAL TO SECTION 3.1

Here, we give the reader insights into the time complexity of Algorithm 1. We estimate the complexity to be $\mathcal{O}(N^2)$, where $N$ is the number of vertices in the graph.

The estimation is derived from the following observations:

**1)** The outer and nested loops of Algorithm 1 (line 1 and 3) are both executed $P$ times, where $P$ is the number of partitions, for a total number of $P^2$ executions.

**2)** Execution time of function $sample$ (line 5) is dominated by function $neighbors(p, op)$ (line 6) and the recursive call to $sample$ (line 12).

**3)** The number of recursive calls to $sample$ depends on the number of vertices to sample $nts = \frac{O \cdot N_p}{P-1}$ and $N_p = \frac{N}{P}$, where $O$ is the overlap. It also depends on $avg\_edges_{hop_x, hop_{x+1}}$, which denotes the average number of edges between hop x and hop x+1, where hop 0 is partition p and hop 1 is a different partition $op$. This is because we sample breadth-first: if there are more than $nts$ edges between $p$ and $op$, no recursive call is needed. Otherwise, $sample$ first selects the direct edges between $p$ and $op$ and then moves on to sample more edges farther away (in terms of hops). The number of recursive steps necessary to reach $nts$ samples can be described as:

$$\frac{nts}{avg\_edges_{hop_x, hop_{x+1}}} =$$

$$\frac{\frac{O \cdot N_p}{P-1}}{avg\_edges_{hop_x, hop_{x+1}}} = \{P - 1 \approx P; N_p = \frac{N}{P}\} =$$

$$\frac{O \cdot N}{P^2 \cdot avg\_edges_{hop_x, hop_{x+1}}}$$

where O is the overlap.

**4)** The function $neighbors$ finds the one-hop neighbors between partitions $p$ and $op$, which means that it has to traverse the vertices in $p$ and determine which of its neighbors belong to $op$. Hence, its worst-case complexity is $\mathcal{O}(N_p \cdot avg\_edges_{p,op})$, where $avg\_edges_{p,op}$ is the average number of direct edges between the vertices in $p$ and $op$, and $N_p$ is the number of vertices in $p$. However, $avg\_edges_{p,op}$ is bounded upwards by the average vertex degree $AVD$ which gives that also complexity is bounded upwards by $\mathcal{O}(N_p \cdot AVD)$.

Putting all the observations together, the complexity of the full algorithm can be estimated as:

$$\mathcal{O}(P^2 \cdot \frac{O \cdot N}{P^2 \cdot avg\_edges_{hop_x, hop_{x+1}}} \cdot N_p \cdot AVD) = \{N_p = \frac{N}{P}\} =$$

$$\mathcal{O}(\frac{O \cdot N^2 \cdot AVD}{P \cdot avg\_edges_{hop_x, hop_{x+1}}}) \rightarrow \mathcal{O}(N^2)$$

assuming that $O$ and $P$ are constants, and that $AVD$ as well as $avg\_edges_{hop_x, hop_{x+1}}$ are fixed properties of the input graph.

Note however that the execution time is sensitive to the property $avg\_edges_{hop_x, hop_{x+1}}$: if the graph is very sparse many recursive calls has to be made to meet the sample number criteria, which increases execution time. For the graph and overlap combinations we evaluate, however, no recursive calls are necessary.

## B   APPENDIX: SUPPLEMENTARY MATERIAL TO SECTION 5.2

A per-epoch comparison of the effect of training GraphSAGE on the Amazon2M dataset, distributed on 20 workers. Figure 6 shows the resulting accuracy for epochs 1 to 80. We consider an overhead of 0%, 0.5%, 1.25%, and 2.5% of the *total* vertices in each partition. We observe a clear and substantial increase in accuracy: a small overhead of 2.5% increases accuracy from 85.55% to 87.62% (at 80 epochs).

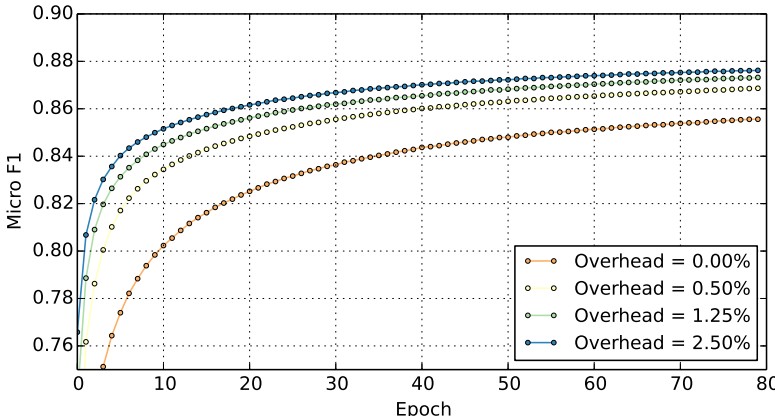

Figure 6: Accuracy per epoch when training GraphSAGE on twenty workers with the Amazon2M dataset, using subgraph approximation.

## C APPENDIX: SUPPLEMENTARY MATERIAL TO SECTION 5.3

This section shows an example of how the number of hidden layers has an impact on the super-linear property we discuss in Section 5.3.

Figure 8 shows an example where Reddit is trained on five machines using GraphSAGE with only one layer (cf. Figure 7 which shows the same situation but for two layers). In comparison to GraphSAGE with two layers, 2% overhead (which is 10% *overlap*) gives a much lower increase in accuracy. In addition, it appears that partitioning affects the one-layer GraphSAGE less than two-layer GraphSAGE, since the one-layer scenario does not really benefit from the super-linearity of including additional vertices.

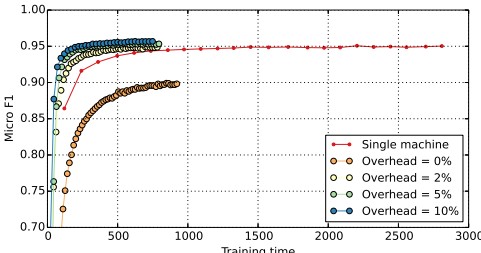 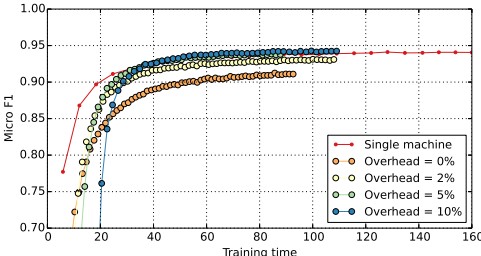

Figure 7: Accuracy vs training time when training two-layer GraphSAGE on five workers with the Reddit dataset.

Figure 8: Accuracy vs training time when training one-layer GraphSAGE on five workers with the Reddit dataset.

