# OpenReview forum: "Distributed Training of Graph Convolutional Networks using Subgraph Approximation"
_ICLR.cc/2021/Conference — Reject_

### Official Review · AnonReviewer2 · 2020-10-13
**Important problem; technical contribution is limited given prior work**

**Rating:** 5
**Confidence:** 4

**Review:**

Graph Convolutional Networks (GCNs) have inspired state-of-the-art methods for learning representations on graphs. However, training GCNs for very large graphs remains an issue because of their memory, computation demands. Towards addressing this issue, the paper proposes a distributed GCN training scheme based on subgraph approximation, and demonstrates its empirical effectiveness on medium-to-large datasets.



## Strengths
1) (Motivation) The paper considers an important and fast-growing topic. Training GCNs for large graphs is important for real-world applications. But, GCNs are known to suffer from large memory footprints.
2) (Motivation) While many previously proposed schemes train GCNs on a single machine, this paper considers a fully distributed setting. As opposed to single-machine GCNs, distributed GCNs do not need the entire graph information for training.
3) (Presentation) The paper is easy to read, and seems well-organised.



## Weaknesses
1) (Significance) An important baseline to compare against is SGC [Simplifying Graph Convolutional Networks, In ICML'19]. SGC is a (non-neural) logistic-regression-based method, of the form $\tilde{X}=A^kX$, that can easily scale to very large graphs. Moreover, since $\tilde{X}$ is pre-computed in SGC, the vertices can be seen as i.i.d instances facilitating distributed training of SGC.
2) (Originality) The novelty of the paper is quite limited as it explores two well-known existing methods viz., GCN [Kipf and Welling, ICLR'17], and GraphSAGE [Hamilton et al., NeurIPS'17] in the distributed setting.  The main modification is the proposed subgraph approximation scheme which seems quite ad-hoc. A lack of theoretical understanding of the approximation scheme limits the contributions.
3) (Quality) The paper can be significantly improved by positioning with and/or exploring many other (more recent) large-scale single-machine graph neural networks (GNNs). It is currently unclear why / whether the proposed subgraph approximation can / cannot be applied to such GNNs for distributed training. Recent publications include (but are not limited to)
i) GraphSAINT: Graph Sampling Based Inductive Learning Method, In ICLR'20,
ii) L^2-GCN: Layer-Wise and Learned Efficient Training of Graph Convolutional Networks, In CVPR'20,
iii) Layer-Dependent Importance Sampling for Training Deep and Large Graph Convolutional Networks, In NeurIPS'19.



## Comments
1) The authors are encouraged to position their work with / compare against the following contemporaneous distributed GNNs in the camera-ready / a future version.
(a) Scaling Graph Neural Networks with Approximate PageRank, In KDD'20.
(b) Reducing Communication in Graph Neural Network Training, In SC'20.
2) The experiments on Reddit and Amazon datasets have demonstrated the effectiveness of the proposed method (which is commendable). Experiments on additional datasets would add great value to the paper. The authors could consider, for example, large-scale datasets considered in publications a) and b) listed above (Microsoft Academic Graph, and Protein graph respectively) for distributed GCN training.



Overall, the motivations are clear and the biggest strengths of the paper. But the major weaknesses especially along the axes of significance, and originality outweigh the strengths.

---

### Official Review · AnonReviewer4 · 2020-10-27
**need more experiment**

**Rating:** 4
**Confidence:** 4

**Review:**

The paper presents a subgraph approximation method to reduce communication in distributed GCN training. The authors observe that up until a certain point it is possible to delete vertices in a graph without a large accuracy loss for GCN training.  Based on the observation, the authors propose the subgraph approximation idea. They claim that if the subgraph on each machine has additional information of 1 or 2 hop neighbors, the accuracy of GCN training can be maintained.

The results are surprising since many information is lost for the border nodes. I am curious how many border nodes are labeled. I suspect that if most of the border nodes are labeled and are used for computing the loss, the model cannot be trained very well.

I don't understand the reason for random sampling in algorithm 1. Since this step is done before the actual training, it is not a stochastic procedure. I suspect the performance will vary in different runs. The authors need to add some explanation and experimental evidence for this design. In my opinion, because different nodes carry different amounts of information, we should select the set of nodes that carry the most information, for example, a node should be added to a partition if it has the most number of connections with the nodes in that partition. I don't see how random selection can perform better than this.

Also, I notice that the authors only tested with GNN with 2 layers in the experiment. Many GNN models have more than 2 layers. I am curious whether the proposed method will still work for GNN with more layers.

My main concern about this paper is its lack of justification. For a heuristic method, I would like to see more experimental results. The authors only experiment with two models on two graphs. It is not convincing that the method is generally applicable to GCN training.

---

### Official Review · AnonReviewer1 · 2020-10-27
**use subgraph aprroximation to compensate missing neigbors**

**Rating:** 4
**Confidence:** 5

**Review:**

This paper studies how to do distributed training for GNNs. For GNNs, nodes are connected so that it is not trivial to do distributed training, because it needs message passing across machines, incurring communication costs. If no message passing across machines, the performance will be bad. To address this problem, this paper stores a subset of the neighbors from other machines to reduce the cost for message passing across machines. The experimental results show improvement to some extent.

However, I have some concerns as follows:

1. In the proposed method, each machine should keep a copy for the subset of neighbor nodes from other machines. Then, the frontier nodes for the subgraph on this machine can aggregate the information from these copied neighbor nodes. However, it is a recursive procedure. These copied nodes should also aggregate their neighbors. How to handle these nodes? Only do aggregation at the end of each epoch? If so, within each epoch, its information is outdated.

2. In the experiment, the used GNNs only have two layers. It is known that GNNs need to recursively aggregate its neighbors. For deeper GNNs, the frontier nodes will aggregate larger-hop neighbors. It is not clear how this method work for deep GNNs.

3. In table 2, for GS, why is the acc of distributed method better than that of a single machine?

4. In this method, there is a critical parameter: overlap. The experiments show how this parameter affect the accuracy. However, a large overlap may include all the immediate neighbors on other machines. If it is the case, the experiments tell us nothing. Thus, it is critical to show the percent of immediate neighbors in the overlap

---

### Official Review · AnonReviewer3 · 2020-10-28
**Basic and effective idea; more work is desired.**

**Rating:** 5
**Confidence:** 4

**Review:**

This paper addresses the problem of training GNNs in a distributed environment (e.g., with multiple machines communicated through a network). In such a setting, the training of GNNs for node classification problems substantially differs from the training of other neural networks, because if the graph and node data are partitioned and distributed across machines, the data held by each machine may not be enough to compute a local gradient.

The proposed solution is simple: augment the data held by each machine in order that they are enough for computing the local gradient. Specifically, the augmented data are nodes that do not belong to the current partition but are within multi-hop neighborhoods of the nodes in this partition. If the augmented data do not all fit in memory (GPU memory in the authors' work), sample the neighborhoods.

This idea is a practical and effective solution. However, the authors' work leaves a lot to be desired. In what follows are questions and suggested work that may help enrich the contribution.

- Section 2.2. It is not entirely clear what "approximated graph" means. Did the authors reduce the training set, cut the graph to a smaller one (but keep the training set), sample multi-hop neighbors for each node in the training set, or do something else? If the approximation means using a smaller training set (which is suggested by the caption of Figure 3), this is not the same as sampling neighbors but keeping the training set. In other words, how does the motivation suggested by Figure 3 support the proposed method?

- Section 3.1. Why breadth-first and uniform sampling? How about nonuniform sampling (e.g., according to node degrees)? How about deterministic sampling (also according to node degrees but done in a deterministic order)?

- Section 3.1. Evenly spreading the overlap across partitions appears too naive. Since the augmentation is done only once at the beginning and does not involve repeated communication, why not adding nodes as many as needed, regardless which partition they come from?

- Section 3 overall. An important point the authors have neglected is load balancing. Since the proposed distributed training is synchronous, load imbalance may be a critical issue. Have the authors considered a more elegant augmentation that encourages load balance, in the sense that each partition may include a different number of additional nodes?

- Section 5.2. The authors mention that the observed speedup (with respect to time) is restricted by ethernet connection, which offers a much smaller bandwidth than infiniband. However, network bandwidth is not the only factor that affects speedup (for example, load imbalance may play a role also). It is important to understand how much each factor contributes. The authors are encouraged to conduct a thorough investigation, convey findings, and design mitigations.

- Figure 5. What does Figure 5 look like if the horizontal axis is epoch? Figure 6 in the appendix partly answers the question, but the "single machine" baseline is missing therein.

- Section 5.2. The authors state "KW-GCN is less resilient to input approximations than GraphSAGE is." Can the authors articulate the reason?

- Section 5.3. What happens if the GNN has more than two layers? Is the proposed method still effective, facing a much larger neighborhood? How would prediction accuracy be affected? The authors are encouraged to conduct investigations.

- Section 5.4. The local time complexity per machine is not the most important factor in distributed computing. Load imbalance, communication, and synchronization costs should all be factored in.

- Additional question to consider for enriching contribution: When a vast amount of computing resources is available (e.g., 100 machines), do the authors recommend using them all, using as few as possible, or somewhere in between? What are the trade offs between overlap size, communication/synchronization cost, and monetary cost?

---

### Decision · Program_Chairs · 2021-01-07
**Final Decision**

**Decision:**

Reject

**Comment:**

The paper proposes a new distributed training method for graph convolutional networks, using subgraph approximation. The reviewers raised multiple challenges, such as novelty, validity of experiments, and some technical issues. The authors did not respond to the reviewers' comments. The AC agreed with the reviewers that the paper, in the current form, is not ready for publication.